# In situ-formable, dynamic crosslinked poly(ethylene glycol) carrier for localized adeno-associated virus infection and reduced off-target effects

Motoi Kato [1,4], Shohei Ishikawa[2,4], Qi Shen[1], Zening Du[1], Takuya Katashima[2], Mitsuru Naito [3], Takao Numahata[1], Mutsumi Okazaki[1], Takamasa Sakai [2✉] & Masakazu Kurita [1✉]

The adeno-associated virus (AAV) is a potent vector for in vivo gene transduction and local therapeutic applications of AAVs, such as for skin ulcers, are expected. Localization of gene expression is important for the safety and efficiency of genetic therapies. We hypothesized that gene expression could be localized by designing biomaterials using poly(ethylene glycol) (PEG) as a carrier. Here we show one of the designed PEG carriers effectively localized gene expression on the ulcer surface and reduced off-target effects in the deep skin layer and the liver, as a representative organ to assess distant off-target effects, using a mouse skin ulcer model. The dissolution dynamics resulted in localization of the AAV gene transduction. The designed PEG carrier may be useful for in vivo gene therapies using AAVs, especially for localized expression.

[1] Department of Plastic and Reconstructive Surgery, Graduate School of Medicine, The University of Tokyo, 7-3-1, Hongo, Bunkyo-ku, Tokyo, Japan. [2] Department of Bioengineering, School of Engineering, The University of Tokyo, 7-3-1, Hongo, Bunkyo-ku, Tokyo, Japan. [3] Center for Disease Biology and Integrative Medicine, Graduate School of Medicine, The University of Tokyo, 7-3-1, Hongo, Bunkyo-ku, Tokyo, Japan. [4]These authors contributed equally: Motoi Kato, Shohei Ishikawa. ✉email: sakai@tetrapod.t.u-tokyo.ac.jp; kuritam-pla@h.u-tokyo.ac.jp

Adeno-associated viruses (AAVs) are powerful vectors for in vivo gene transduction. AAVs are clinically used as replacement therapies to treat conditions such as lipoprotein lipase deficiency, spinal muscular atrophy, retinal dystrophy, and hemophilia[1]. Recently, possible applications of AAVs have been expanded to include localized morbidities[2].

As a result of improvements in manufacturing, the average doses of AAVs used in clinical trials are increasing. This allows the use of higher doses resulting in stronger phenotypes; however, most of the AAV vectors end up in the liver and can cause toxicity there and elsewhere[3]. Thus, controlling the specificity of the gene expression—minimizing off-target effects—is highly important in the development of therapies using AVVs. To date, the specificity has been controlled by the selection and engineering of AAV capsid tissue/cell tropism[4], use of tissue-specific promotors[5,6], and the route of administration[3].

Non-healing wounds, such as wounds resulting from pressure ulcers, diabetic ulcers, or peripheral vascular insufficiency, are of concern in societies with progressively aging populations, and AAV-based in vivo gene transfer is expected to be a useful therapeutic modality[7,8]. Recently, we demonstrated that AAV-mediated in vivo induction of the reprogramming of wound-resident mesenchymal cells toward epithelial cells enabled de novo epithelialization from the surface of ulcers. These results constituted an initial proof of principle for the future development of innovative therapies[9,10].

In the search for clinically applicable methods that can improve the safety of AAV-mediated gene transfer on the surface of cutaneous ulcers, we investigated the development of materials for the controlled release of AAVs on the surface of ulcers[11], using poly(ethylene glycol) (PEG) as a carrier. PEG, which is widely used in biomaterials because of its bio-inert nature, functions as a carrier by encapsulating or immobilizing a drug or virus in vesicles, and hydrogels[12,13]. Injectable hydrogel systems that can be administered minimally invasively at a desired site have received increasing attention for controlled virus delivery because of their ability to change from a highly viscous matrix to a low viscosity liquid[14,15]. However, injection of low viscosity liquid hydrogels can cause diffusion of the encapsulated virus, that is, rapid burst release from the injection site, which potentially impairs the treatment of localized virus infections.

We hypothesized that an AAV encapsulated in a PEG carrier with an appropriate degradability would undergo localized release at the surface of cutaneous ulcers. We designed a PEG carrier consisting of a dynamically crosslinked polymer network (hereafter referred to as PEG slime) that enhanced the specificity of gene transduction to the surface of ulcers with reduced local or distant off-target effects. The use of highly viscous PEG slime demonstrates a new method for highly localized gene transduction.

## Results

### Design and characterization of PEG carriers: hydrogel, sponge, and slime

We prepared three types of PEG matrices as possible carriers for AAVs with different crosslinking and pore sizes. All the PEGs were formed by similar protocols; that is, dissolving mutually cross-linkable tetra-functional PEGs and mixing equal volumes of the same concentrations of the PEG precursors (Fig. 1a). (1) PEG hydrogel[16] (Fig. 1b): a synthetic hydrogel that acts as an artificial extracellular matrix was formed by the crosslinking of propylamine-terminated tetra-armed PEG (tetra-PEG-PA) and succinimidylglutarate-terminated tetra-armed PEG (tetra-PEG-GS) via amide linkages, and the ester bonds in tetra-PEG-GS enabled the hydrogel to be hydrolysable. (2) PEG sponge[17] (Fig. 1c): a porous PEG hydrogel in which a phase-separated structure was achieved by adding potassium sulfate to the precursor solutions. (3) PEG slime[18] (Fig. 1d): a viscoelastic polymer liquid prepared by mixing D(+)-glucono-1,5-lactone-terminated tetra-armed PEG (tetra-PEG-GDL) and 4-carboxy-3-fluorophenylboronic acid-terminated tetra-armed PEG (tetra-PEG-FPBA) to form reversible and fluctuating crosslinking via cyclic ester bonds.

Despite the similarities in the synthetic protocols, these PEG carriers had distinct characteristics and structures. The difference in the mechanical properties was macroscopically observed under compression: the PEG hydrogel and sponge showed completely elastic deformation, while the PEG slime showed a gradual loss in the original shape over time. In gross observations, a PEG slime prepared in a spherical shape could maintain the shape when grasped with tweezers, similar to a hydrogel soon after grasping. However, the grasped slime dropped later as a highly viscous liquid from the tweezers and eventually became a flat shape on the table (Fig. 1e). The difference was then quantified by stress relaxation testing, which observes the time-dependence of Young's modulus [$E(t)$] after imposing a compression to the carrier. With time evolution, the $E(t)$ value drastically decreased for the PEG slime, slightly decreased for the PEG sponge because of the extraction of water, and was constant for the PEG hydrogel (Fig. 1f). Fluorescence recovery after photobleaching (FRAP) measurements (Supplementary Fig. 1) showed that the fluorescence intensity of PEG slime was recovered, indicating that stress relaxation originated in the dynamic covalent bonds between GDL and FPBA. The fluorescence intensity of PEG slime was completely recovered ~60 s after photobleaching, whereas that of PEG hydrogel was unchanged. This confirmed the translational movement of the constitutive molecules of PEG slime and the dynamic crosslinks between these molecules. Microscopic observations of the PEG carriers prepared with red fluorescence-labeled PEG precursors (the terminal moiety of the PEG precursors was partly modified with red fluorescent dye as described previously[17]) indicated structural differences between the carriers. In the PEG sponge, a sea-island structure where black cavities were dispersed in the image, with pore sizes in the order of magnitude of 10 μm, was observed. In the PEG hydrogel and slime, no characteristic structure was observed, suggesting that the network size was in the order of magnitude of nm (Fig. 1g). Phase separation was induced by the addition of an inorganic salt, which decreased the solubility of the PEG, resulting in an opaque hydrogel. During the gelation process, the PEG precursors connect with each other, increasing the molecular weight, which drives the phase separation[17]. The salt concentration was selected to induce the phase separation during the gelation and not in the original precursor solutions[17].

### In vitro evaluation of PEG carrier characteristics

To investigate the differences in the physical characteristic of the PEG carriers in detail, we performed in vitro comparative analyses. An experimental model was designed to facilitate understanding of the dissolution kinetics. Red fluorescence-labeled PEG carriers were prepared on culture inserts with a pore size of 8.0 μm (i.e., smaller materials could pass through) (Fig. 2a). For the detailed assessment of PEG slimes, we prepared a PEG slime that was designed to dissolve faster compared with the standard PEG slime (referred to as PEG soft-slime). The dissolution rate, calculated by measuring the fluorescence intensity of the eluted samples passed through the insert at predetermined time intervals, was gradually increased for each scaffold, and the dissolution rates at 24 h ($D_{sample}$) were $D_{hydrogel} = 19\%$, $D_{sponge} = 16\%$, $D_{slime} = 86\%$, and $D_{soft-slime} = 90\%$ (Fig. 2b). Furthermore, we encapsulated fluorescence silica nanoparticles with a diameter of 30 nm as model materials similar in

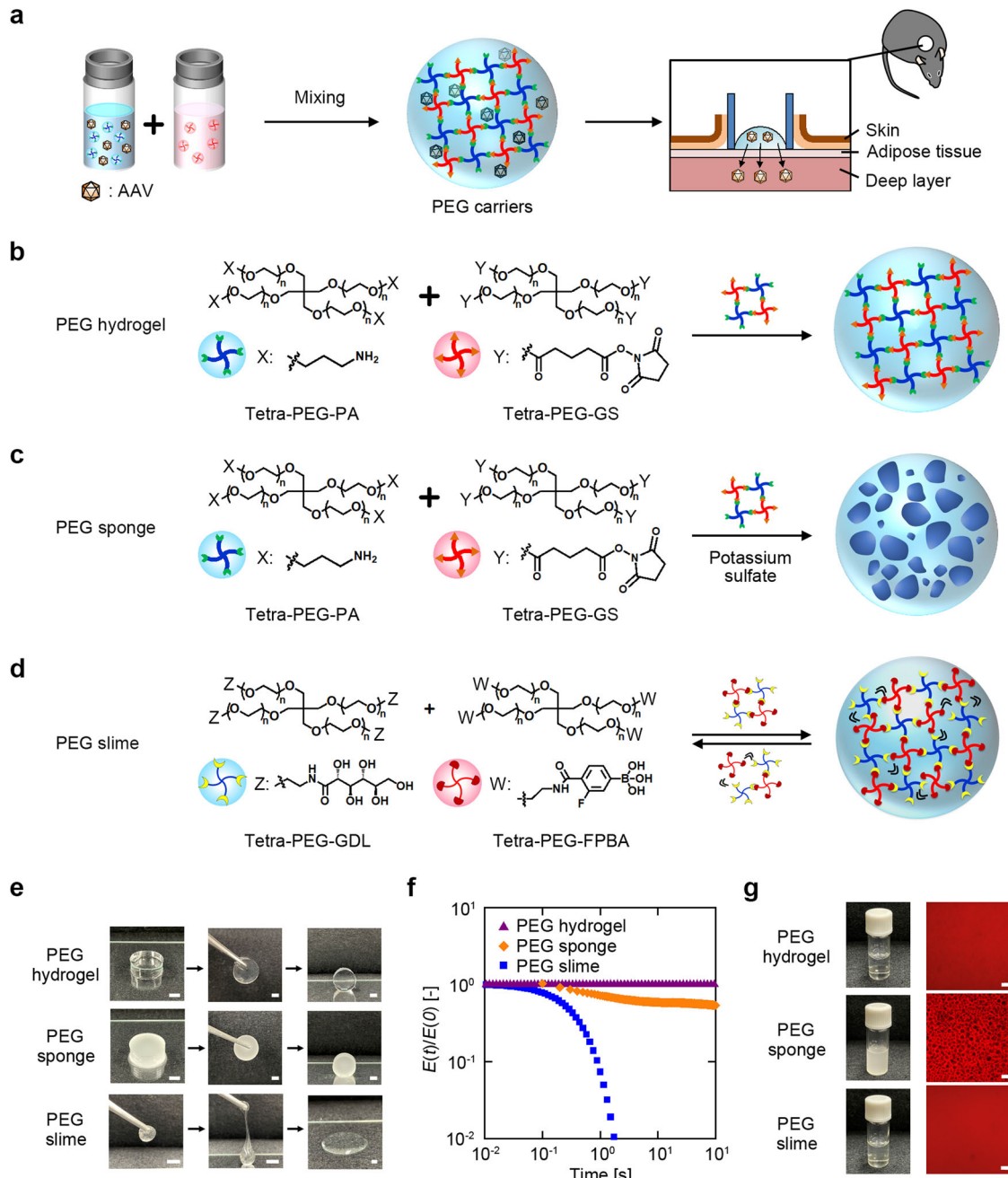

**Fig. 1 Conceptual scheme of AAVs encapsulated in PEG carriers and their characteristics. a** Schematic illustration showing the in situ process for AAV administration to a skin ulcer surface using PEG carriers. Tetra-PEGs, which were functionalized with mutually cross-linkable groups and dissolved with PB containing AAV, were mixed in equal volumes, resulting in the formation of PEG carriers encapsulating the AAV. The prepared PEG carriers were administered to the surface of skin ulcers on mice backs, and AAV was diffused by the dissolution of the PEG carriers. **b–d** Preparation of PEG carriers. PEG hydrogel (**b**) prepared with tetra-PEG-PA and tetra-PEG-GS, PEG sponge (**c**) prepared with the same precursors as the PEG hydrogel, except potassium sulfate in PB was added to create a phase-separated structure. PEG slime (**d**) was prepared from tetra-PEG-GDL and tetra-PEG-FPBA to obtain a dynamic crosslinked system. **e** Photographs of the PEG hydrogel, PEG sponge, and PEG slime, showing the viscoelastic behavior on being grasped. Scale bars represent 5 mm. **f** Young's modulus ($E$) as a function of time ($t$) of the PEG hydrogel (purple triangles), PEG sponge (orange diamonds), and PEG slime (blue squares). $E(t)$ was normalized by $E$ at 0 s [$E(0)$]. **g** Photographs and confocal laser scanning microscope images of PEG carriers in the as-prepared state. Scale bars represent 100 μm.

diameter to AAVs, and investigated the release profile of the fluorescence silica nanoparticles from the PEG carriers. The release behavior of the particles encapsulated in each carrier showed a similar trend to that of the dissolution behavior, and the release rates at 24 h ($R_{sample}$) were $R_{hydrogel} = 38\%$, $R_{sponge} = 33\%$, $R_{slime} = 75\%$, and $R_{soft-slime} = 92\%$, indicating that the release of the nanoparticles was governed by the dissolution of the carriers

(Fig. 2b). A plot of the dissolution rate versus the release rate (Fig. 2c) was linear for the PEG slimes. This strong correlation demonstrated that the release of the particles was governed by the dissolution of the PEG slimes. This result indicated that the PEG slimes were pseudo-solids, where the diffusion of the particles was highly limited, but the cargoes could be eluted to the solution from the liquid form.

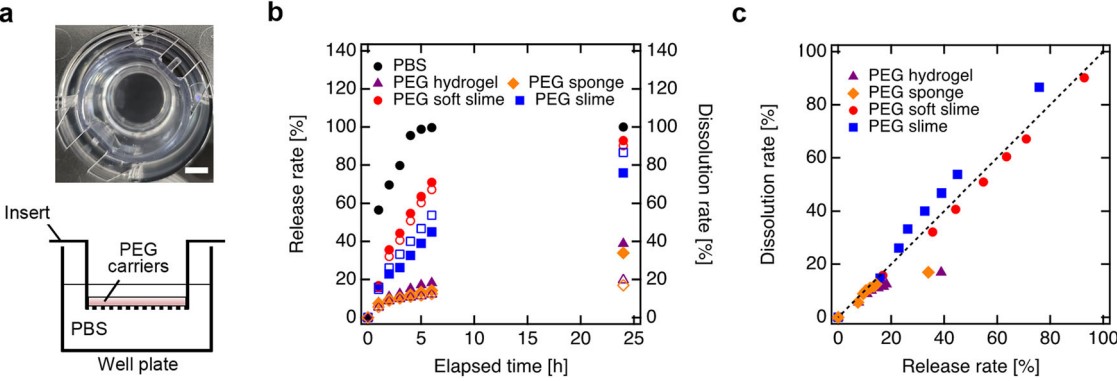

**Fig. 2 In vitro evaluation of PEG carriers: dissolution and diffusion of silica nanoparticles as model viruses. a** Schematic illustration showing the in vitro dissolution assessment. Photographs show the upper view of insert. Scale bars represent 5 mm. **b** Time evaluation of dissolution (open symbols) and release (closed symbols) rate for PEG hydrogel (purple triangles), PEG sponge (orange diamonds), PEG slime (blue squares), and PEG soft-slime (red circles) prepared on a culture insert placed on a culture plate. **c** Relationship of dissolution rate and release rate.

Notably, the release rates for the PEG slimes were almost constant for 7 h; that is, ideal zero-order release kinetics were achieved. This result was because the elution, which governs the particle release, was approximated as zero-order as it occurred only through the membrane. In contrast, the release of the particles from the PEG hydrogel and sponge did not correlate with a high release rate region. This result was because these carriers were solid, and the particles were mainly released by diffusion.

**Efficiency of AAV gene transduction in skin ulcers**. For therapeutics for wounds, gene transduction methods that are highly specific to the surface of ulcers are preferred. To investigate the effects of the PEG carriers on gene transduction with an AAV for the treatment of skin ulcers, isolated skin ulcers that simulated the central portion of a large cutaneous ulcer were employed[10]. We surgically removed skin from the back of mice to generate an ulcer and isolated the resulting wound from the surrounding skin using a silicone chamber sutured to the deep fascia[19,20] (Supplementary Fig. 2a). By isolating the wound from the surrounding skin using the chamber, we can study phenomena on the ulcer surface without interference from the surrounding skin, such as epithelialization and wound contraction. GFPNLS-AAVDJ [a type of AAV optimized for cells in skin ulcers[10] encoding green-fluorescent protein (GFP) with a nuclear localization signal (NLS)] was mixed with the PEG carriers and inoculated onto the ulcer. We employed the number of positive cells rather than the percentile because we considered that the former would be more appropriate for histological analyses of skin ulcer tissues, which consisted of heterogeneous types of cells with undefined margins. We tested different titers of the virus in PBS (Supplementary Fig. 2b) and observed the ulcer surface with a fluorescence microscope over time (up to day 13) (Supplementary Fig. 2c) and ulcers 72-h after administration of $10^{10}$ Gene copies (GC) of virus were selected for quantitative analysis (Fig. 3a). Observations of the ulcer surfaces with a fluorescent stereoscope indicated that GFPNLS-positive cells were more frequently observed in animals treated with AAVs encapsulated in PEG slime, PEG soft-slime, or phosphate-buffered saline (PBS) than those in PEG hydrogel or PEG sponge (Fig. 3b, c).

**Superficial layer-specific AAV gene transduction on skin ulcers**. While PEG hydrogel and PEG sponge encapsulation reduced the gene transduction efficiency on the surface of skin ulcers, PEG slime encapsulation retained the original transduction efficiency as observed with PBS. The PEG slimes were used

for further histological evaluation with emphasis on the gene transduction specificity in the superficial layer of skin ulcers. Ulcers treated with GFPNLS-AAVDJs formulated with PEG slime, PEG soft-slime, and PBS were prepared and histologically analyzed for the number of GFPNLS-positive cells in the superficial (granulation and adipose tissue) and deep (fascia and muscle) layers (Fig. 3d and Supplementary Fig. 2a). In agreement with the results of the stereoscopic assessment, no significant differences were found in the number of positive cells in the superficial layers of animals treated with PEG slime, PEG soft-slime, or PBS. In contrast, the number of GFPNLS-positive cells in the deep layers decreased significantly in the PEG slime-treated animals, compared with PBS-treated mice, and the degree of reduction was greater in the PEG slime-treated animals, compared with the PEG soft-slime-treated mice (Fig. 3e). To further investigate the possible differences in gene transduction efficiency in terms of the multiplicity of gene transduction, mixtures of two different fluorescent colored AAVs (GFPNLS-AAVDJ and mCherryNLS-AAVDJ) were administered on the surface of the ulcer, and the frequency of duplicated gene transduction was measured (Supplementary Fig. 3a). The number of duplicated gene transduction was consistent between PBS and PEG slime, further indicating that there were no significant differences in mice treated with PEG slime and PBS in terms of the multiplicity of gene transduction (Supplementary Fig. 3b). Because the use of a PEG slime carrier improved the local, layer-specific-efficiency of the gene transduction, compared with the other carriers, we further investigated the distant off-target effects of this carrier.

**Reduction of distant off-target gene expression in the liver**. AAV cell tropism differs among serotypes, and thus the importance of each organ as the distant off-target depends on the serotype. Previously, we assessed the distribution of AAVDJ in various organ tissues following the injection of luciferase-AAVDJs and found that luciferase expression is primarily confined to the liver[10]. To investigate the possible influence of PEG slime encapsulation, off-target GFPNLS expression in the liver was evaluated 28 days after inoculation of the ulcers with the AAVDJ vector in a PEG slime carrier, or PBS, in a skin chamber. A small number of GFPNLS-positive cells were observed on the surface of the liver using a stereoscope (Fig. 4a). Quantitative comparison using qPCR of the AAV titers in the liver indicated that the genomic copies were significantly reduced with the use of the PEG slime carrier, compared with PBS (Fig. 4b). Thus, the use of a PEG slime carrier has the potential to reduce distant off-target effects.

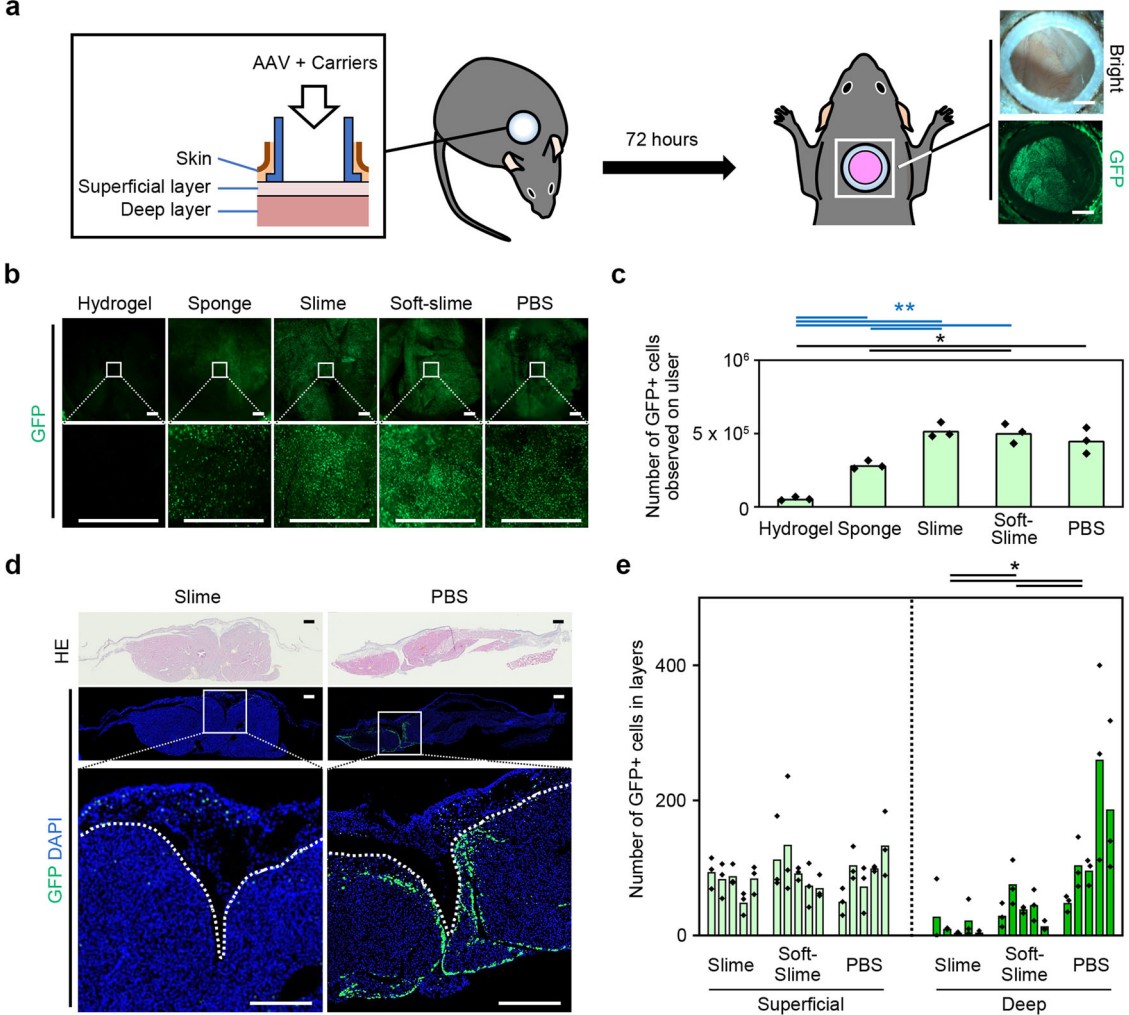

**Fig. 3 Infection ability of AAV diffused from PEG carriers. a** Administration of GFPNLS-AAVDJ (AAV-DJ serotype, $1 \times 10^{10}$ GC per mouse) encapsulated in PEG carriers to a raw surface created in 1 cm diameter silicon chamber attached on mice backs. **b** Representative GFP-positive cells on mice back skin ulcers 72 h after administration, observed under a stereoscopic microscope. Bar = 2 mm. **c** Numbers of superficially expressed GFP-positive cells, 72 h after the administration of each carrier containing AAV. The overlaid dot plot indicates the distribution of the data. A significantly higher number of superficial GFP-positive cells were observed with PEG slime or soft-slime, compared with the other PEG carriers. No significant difference was detected between PEG slime or soft-slime and PBS. * < 0.05, ** < 0.01 (n = 3). **d** Representative slices showing AAV-GFPNLS infection 72 h after administration, of PEG slime and PBS formulations. Many GFP-positive cells were observed in the deep layer when AAV-GFPNLS was administered with PBS, whereas most GFP-positive cells were found in the superficial layer after administration with PEG slime. The dotted lines indicate the margin between the superficial and deep layers. Bar = 500 µm. **e** Number of GFP-positive cells in the superficial layer (above the subcutaneous tissue) and deep layer (to muscular tissue) in each animal. The overlaid dot plot indicates the distribution of the data. Note that the PEG slime prevented deep tissue infection significantly, but there were no significant differences between the formulations in the number of GFP-positive cells in the superficial layer. * < 0.05 (n = 5).

**Behavior of silica nanoparticles on the surface of skin ulcers**. Through a series of in vivo experiments, PEG slimes were found to be useful carriers for surface-specific AAV gene transduction, while PEG sponge and PEG hydrogel carriers reduced the gene transduction efficiency, compared with PBS. We believe that these differences were related to the release profiles of the PEG slimes as determined in the in vitro assessments described above. To elucidate the possible mechanistic properties specific to the PEG slime, fluorescently labeled AAVs[21–23] were tracked, and we found that reliable detection was difficult because of the weak fluorescence signals relative to the background signals (Supplementary Fig. 4a–c). Instead, the behavior of silica nanoparticles on the surface of skin ulcers was investigated. In the initial trial, green-fluorescent silica nanoparticles (with a diameter of 30 nm, similar to that of AAVs[24]) in PBS were inoculated. The nanoparticles were localized in the superficial layer at first and then diffused into the deep layers, such as adipose and muscle tissues, with time. Importantly, only a few particles remained on the ulcer surface at 24 h (Supplementary Fig. 5). Accordingly, 24 h was used as the experimental time frame for the following quantitative analysis of the influence of the PEG carrier on the dissolution of PEG and the release of the silica nanoparticles.

Green-fluorescent silica nanoparticles were mixed with PEG carriers or PBS, and then inoculated on an ulcer (Fig. 5a). With the PEG carriers, more silica nanoparticles remained on the superficial layer at 24 h, compared with PBS. This effect was greater for the PEG slime and PEG soft-slime, than with the PEG hydrogel and PEG sponge (Fig. 5b, c). The prolonged surface-specific distribution of silica nanoparticles using a PEG slime carrier might be associated with surface-specific AAV gene transduction.

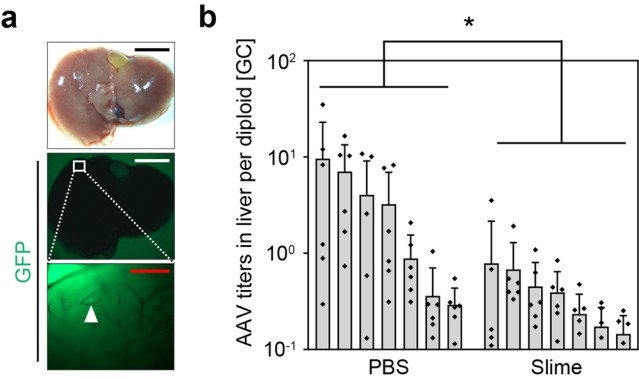

**Fig. 4 Reduction of off-site effects in liver. a** GFPNLS-AAVDJ (AAV-DJ serotype, $5 \times 10^{10}$ GC per mouse) encapsulated in PEG slime or diluted with PBS were administered to a raw surface created on mice backs. After 4 weeks, the liver was harvested for quantitative analysis, including qPCR assessments. There were insufficient GFP-positive cells for statistical analysis. The filled arrow head indicates a GFP-positive cell observed in the liver using a stereoscope. Black and white bar = 5 mm, Red bar = 500 μm. **b** Genomic DNA was extracted from six lobes of each harvested liver to assess AAV infection rates. The AAV titers were significantly decreased in the slime group. ($n = 7$) * <0.05.

**AAVs decay over time**. With PEG slime, the number of silica nanoparticles in the superficial layer increased, although AAV-mediated gene transduction in the superficial layer did not increase. We considered that this discrepancy might result from the decay of AAVs in PEG over time.

To determine the possible effects of storage time at body temperature and solvent solution on the gene transducing ability of AAVs, we tested the gene transducing ability of AAVs kept for 0, 3, 6, 12, 18, and 24 h at 37 °C with or without mouse blood serum in vitro (Fig. 6a). When AAVs were kept at 37 °C for 3 to 18 h in the absence of serum, the gene transducing ability of AAVs gradually reduced to one tenth of the original ability. In contrast, when AAVs were kept for 24 h with serum, the gene transducing ability of AAVs was maintained (Fig. 6b). The gene transducing ability of AAVs within PEG slime, formulated with PEG and PBS, might reduce over time, whereas this reduction might be attenuated once AAVs were released from PEG slime. Over time, the decay of AAVs in PEG might contribute to the discrepancy in the silica nanoparticle behavior and gene transduction efficiency in the superficial layer.

## Discussion

In the present study, a variety of PEG carriers were assessed using an in vivo skin ulcer model. The PEG slime successfully localized gene expression on the ulcer surface, with a significant reduction in gene expression in both the deep muscle layer and the liver, compared with the other carriers investigated.

To investigate the detailed kinetics of AAVs, fluorescent labeling was used. However, reliable tracing was difficult because of the weak fluorescence signals relative to the background signals. Alternatively, we employed silica nanoparticle because each particle exhibited strong fluorescence and could be reliably traced in skin ulcer tissue samples. The possibility that AAV and silica nanoparticles may not exhibit the same behavior is a fundamental limitation of the current study.

The in vivo gene transduction efficiency differed between the different PEG carriers. This difference was attributed to the dissolution mechanisms resulting from the different crosslinking forms. In the PEG hydrogels and sponges, all the molecules were connected through irreversible bonds. Thus, these carriers had

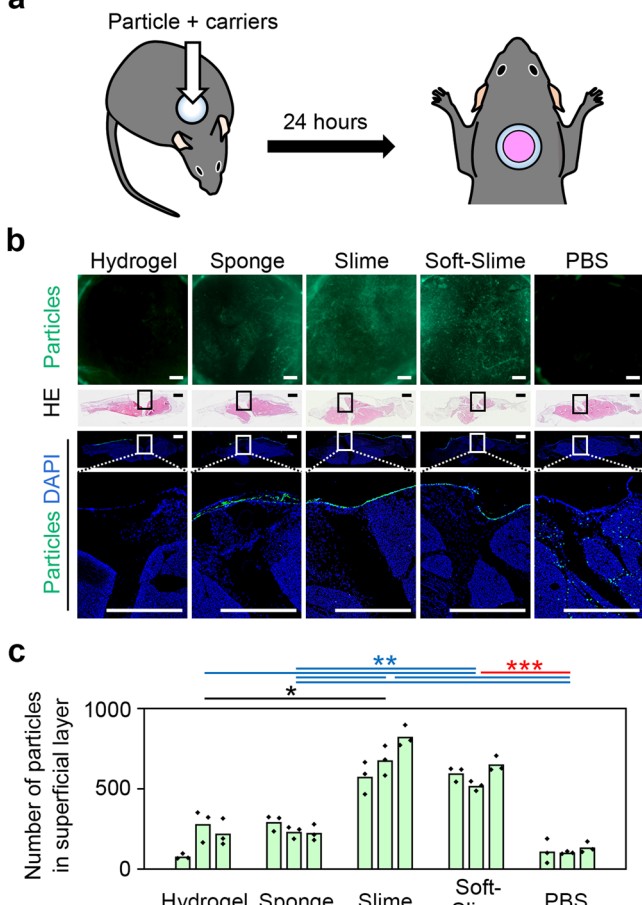

**Fig. 5 Dispersion behavior of silica nanoparticles diffused from PEG carriers. a** Administration of fluorescent silica nanoparticles encapsulated in PEG carriers to the raw surface created on mice backs. PEG hydrogel and sponge carriers were injected into the chamber, while PEG slime was inserted as the slime was unable to be injected. The roof of the set silicon chamber was opened for observation 24 h after administration. **b** Comparison of particle diffusion from the PEG carriers, 24 h after the administration. Representative images from a stereoscope and slices. Note, the particles administered with PEG slime were concentrated on the ulcer surface, whereas particles were diffused into muscles when administered with PBS. Bar = 1 mm. **c** The number of fluorescent silica nanoparticles in the superficial layer. The overlaid dot plot indicates the distribution of the data. With PEG slime and soft-slime, the number of fluorescent nanoparticles in superficial layer increased significantly. ($n = 3$) * <0.05, ** <0.01, *** <0.001.

properties characteristic of solids; in vitro characterization showed that they exhibited a finite elastic modulus under constant strain (Fig. 1f). In vitro assessment of the PEG dissolution rates and particle release rates showed that there was a significant reduction in both rates when hydrogels and sponges were used, compared with the other carriers, which suggested that the nanoparticles were trapped in the networks (Fig. 2b).

In general, solid polymeric materials, such as PEG hydrogels and sponges, are considered to degrade homogeneously, which is called bulk degradation, rather than to degrade gradually from the surface[25–27]. In other words, chain scission homogeneously occurs over time by hydrolysis. Eventually, when the network connectivity falls below a certain level, the PEG hydrogels and sponges transition from a solid to a liquid. Therefore, PEG hydrogels and sponges are thought to have reduced particle

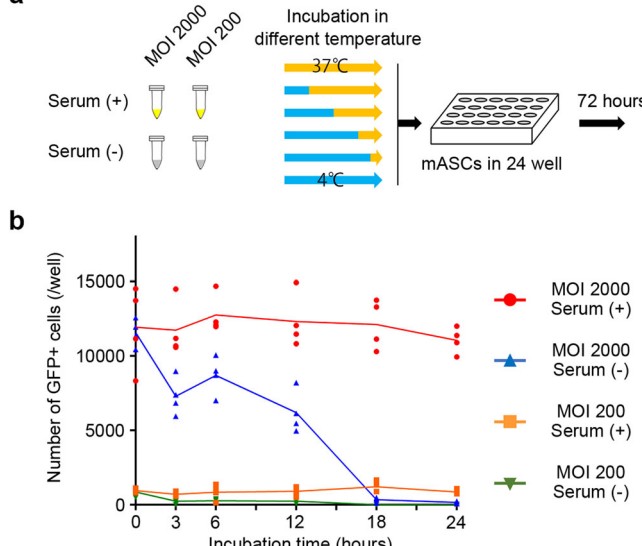

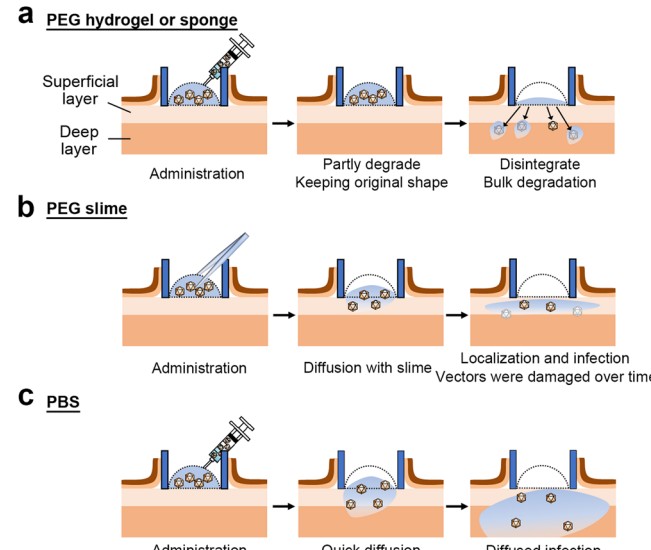

**Fig. 6 AAVs decay over time. a** Gene transduction efficiency of GFPNLS-AAVDJ kept for 0, 3, 6, 12, 18, and 24 h at 37 °C with or without mouse blood serum was tested on mouse adipose-derived stromal cells (mASCs) in vitro. **b** The gene transducing ability of GFPNL-AAV gradually reduced after storage at 37 °C in the absence of serum, while it was consistent after storage for 24 h with serum. The overlaid dot plot indicates the distribution of the data. ($n = 4$).

**Fig. 7 Schematic illustration of the localized virus infection mechanism using PEG slime. a** PEG hydrogel or sponge carriers encapsulating AAVs were prepared by mixing gelling precursors of tetra-PEGs bearing succinimidylglutarate (tetra-PEG-GS) and propylamine (tetra-PEG-PA) groups at the ends, which were mutually crosslinked by condensation to form amide linkages, which are irreversible bonds. Ester groups originally introduced to tetra-PEG-GA were introduced between the PEG molecules, which are eventually hydrolyzed to degrade the hydrogel or sponge. The degradation process can proceed homogeneously in these carriers, called bulk degradation, rather than degradation proceeding gradually from the surface. As the carriers are degraded, encapsulated AAVs diffuse to the superficial or deep layers and infect the surrounding cells. **b** PEG slime was prepared by mixing aqueous solutions of diol- or boronic acid-terminated tetra-PEGs (tetra-PEG-GDL and tetra-PEG FPBA), which were mutually crosslinked by a cyclic ester reaction to form coordination bonds, which are reversible bonds. The reversible crosslinking endowed the PEG matrix with a highly viscoelastic property, making the matrix a liquid. This PEG liquid could be varied in shape to cover a skin ulcer and encapsulates AAVs without diffusing through the skin layers. PEG slime diffuses into the superficial layer, rather than being degraded, from the interface between the slime and the skin ulcer. As PEG slime can penetrate into the superficial layer, the PEG slime is swollen and diluted, leading to infection in the surrounding cells, which results in reducing distant off-target gene expression in the liver. **c** AAV administered with PBS diffused quickly. The gene transduction potency is not damaged until AAV reaches deep layers. Therefore, AAV effectively infect both superficial and deep layers.

release, compared with the other carriers, because they penetrate into tissues with trapped AAV particles while being degraded (Fig. 7a). Fluorescent particle experiments to evaluate the physical movement of nanoparticles showed that both hydrogels and sponges resulted in a reduced number of particles in the superficial layer of ulcers (Fig. 5b, c), compared with PEG slimes.

AAV particles require direct contact with cells for successful infection by endocytosis[6]. Therefore, AAV infection was suppressed when AAV particles were trapped in a PEG network because there was no direct contact with cells. In other words, unreleased AAV particles are not infectious, which resulted in decreased GFP expression on the ulcer surface after AAV administration using a PEG hydrogel or sponge.

In contrast, the molecules were connected through reversible bonds in PEG slime, resulting in an equilibrium between the binding and dissociation of the molecules (Supplementary Fig. 1). The elastic modulus of the slime decreased and reached zero over time (Fig. 1f), indicating that the slime was a liquid from a thermodynamic point of view. Therefore, PEG slime applied to the ulcer surface could diffuse as a liquid and permeate from the ulcer surface into the deep tissues; PEG slime and PBS were considered equivalent in terms of carrier and AAV diffusion.

PEG slime showed notable characteristics that differed from PBS. Both the dissolution and release rates of slime were lower than those using PBS (Fig. 2b), and the dissolution and release rates over time were almost identical (Fig. 2c). These findings indicated that the particles were released at the same time as the local dissociation of PEG in the slime. The nanoparticles contained in the PEG slime were confined to the ulcer surface after application (Fig. 5b), limiting penetration from the interstitium to the deeper tissues, which may be because the slime remained on the ulcer surface longer than did PBS. In other words, unlike the solid-like PEG hydrogels and sponges, the particles in the liquid-like PEG slime diffused translationally[27], and AAV infection occurred predominantly on the ulcer surface where the PEG slime was located (Fig. 7b).

The PEG slime was more localized during viral infection than PEG soft-slime, which had lower viscoelasticity, resulting in predominantly reduced deep infection using PEG slime (Fig. 3e). Because the local off-target deep layer infections were limited, the off-target expression in the liver was also decreased.

In previous reports, gene therapy with viral vectors using carriers has resulted in a suppressed viral vector release rate from the carriers. For example, a lentivirus with a hydrogel and analogs resulted in prolonging the local activity in vitro[27], and in vivo[28]; an adenovirus carried with fibrin had increased bioactivity and a prolonged half-life in situ[29]; and an AAV with gelatin had an increased local concentration of the bioactive substance[30].

The behavior of silica nanoparticles on the surface of skin ulcers with or without PEG carriers indicated the potential utility of PEG carriers in the delivery of other types of drugs and biomolecules. Elucidation of other behaviors, such as dose dependency along with time-dependent diffusion, biodistribution, and

clearance of subcutaneously administered substances should be performed in future studies.

PEG slime might increase the absolute amount of AAV particle administered in the superficial layer, although this increase was canceled by the decay of AAVs within PEG over time, thereby reducing the number of positive cells in deeper layers as well as the total number of positive cells. As a consequence, the tetra-PEG carrier developed in the present study did not significantly improve the local activity. Gene transduction was localized in the superficial layer of the skin and there was significant attenuation of off-target effects.

We have described a new technology to generate expandable epithelial tissues via the direct reprogramming of wound-resident mesenchymal cells, which enables all regions of the wound to re-epithelialize without the spatial constraints observed during normal healing[9,10]. Transduction of reprogramming factors potentially induces direct reprogramming of non-epithelial cells to epithelial cells and hence the formation of ectopic epithelial tissues at sites other than the superficial layer of skin ulcers. In addition, it is desirable to minimize potential adverse reactions that the transgene may induce in remote organs.

In our initial proof-of-concept study, no adverse reactions were detected in the limited number of small animals. Nevertheless, all possible measures should be made for the further development toward clinical applications. We consider the current findings might be insightful not only for the development of gene therapy toward skin ulcers, but also for any types of therapeutic developments that are localized and thus expected to have extremely powerful effects.

The other limitations of the current study lie in the differences between human and mouse skin structures[31]. The major differences include the thickness of layered components, such as the dermis and subcutaneous adipose tissue, the absence of sweat glands in mice, and the presence of the thin muscular layer known as *Panniculus carnosus* in mice back skin. These differences might not substantially influence the findings of the current study because the experiments were performed with the skin ulcer in a silicone chamber. Nevertheless, careful interpretation of the results is required during clinical applications. The other limitation lies in the experimental model. Administration of liquid AAVs in a closed chamber is completely different from actual administrations in clinical settings, which involve direct application of the solution to the open wound surface. Use of viscoelastic liquids as a carrier of AAVs might be advantageous for reducing AAV shedding[32], although it could not be determined experimentally.

The use of PEG slime, which is a highly viscoelastic liquid, as a carrier for AAVs resulted in localized gene expression in the superficial layer of the skin ulcer and reduced the off-target effects, compared with other carriers and a PBS control. The controlled release using the PEG carrier developed in the present study paves the way for future localized gene therapies.

## Methods

**Materials**. The tetra-PEG-PA and tetra-PEG-GS ($M_w = 20$ kg mol$^{-1}$) (NOF Corporation, Japan) were used without further purification. Poly(ethylene glycol) functionalized with primary amine ($M_w = 20$ kg mol$^{-1}$) (tetra-PEG-NH$_2$) (SINO-PEG, China) was used without further purification. Potassium sulfate (K$_2$SO$_4$), super-dehydrated methanol, super-dehydrated dimethyl sulfoxide, GDL, FPBA, triethylamine (TEA), 4-(4,6-dimethoxy-1,3,5-triazin-2-yl)-4-methylmorpholinium chloride $n$-hydrate (DMT-MM), and 4% paraformaldehyde phosphate buffer solution (PFA) (FUJIFILM Wako Pure Chemical Corporation, Japan) were used without further purification. Alexa Fluor$^{TM}$ 594 NHS ester (succinimidyl ester) (Alexa-NHS) and Dulbecco's phosphate-buffered saline (DPBS) (Thermo Fisher Scientific, USA) were used without further purification. Phosphate buffer (pH 7.4; PB) was prepared at a concentration of 200 mM. RC15-AC (Sartorius, Germany), ARVO$^{™}$ X3 (PerkinElmer, USA), and LSM 800 (ZEISS, Germany) were used as the 0.22-µm filter, microplate reader, and confocal laser scanning microscope (CLSM),

respectively, in all experiments. Milli-Q water was used as the water throughout the study.

**Preparation of diol or boronic acid-terminated tetra-PEG (tetra-PEG-GDL and tetra-PEG-FPBA)**. Tetra-PEG-GDL and tetra-PEG-FPBA were synthesized by coupling the relevant molecules to tetra-PEG-NH$_2$ according to a previous report[18]. For tetra-PEG-GDL, tetra-PEG-NH$_2$ (1000 mg, 0.05 mmol), GDL (89 mg, 0.5 mmol, 10 equiv. of tetra-PEG-NH$_2$), and TEA (50 mg, 0.5 mmol, 10 equiv. of tetra-PEG-NH$_2$) were dissolved in 30 ml of super-dehydrated methanol. The mixture was stirred and incubated for 3 days at 35 °C. The mixed solution was then dialyzed against excess methanol and water for 24 h each using a Spectra/Por® dialysis membrane (MWCO: 6000–8000 Da, Spectrum Laboratories, Greece). The solution was filtered with a 0.45-µm filter and freeze-dried to obtain tetra-PEG-GDL (yield: 900 mg). The obtained polymer was characterized by $^1$H-NMR (Bruker Avance DPX-400 MHz, USA) using D$_2$O as the solvent.

For the synthesis of tetra-PEG-FPBA, tetra-PEG-NH$_2$ (1000 mg, 0.05 mmol), FPBA (91 mg, 0.5 mmol, 10 equiv. of tetra-PEG-NH$_2$), and DMT-MM (138 mg, 0.5 mmol, 10 equiv. of tetra-PEG-NH$_2$) were dissolved in 30 ml of super-dehydrated methanol. The mixture was stirred and incubated for 3 days at 35 °C. The mixed solution was then dialyzed with against excess 10 mM HCl aq., 10 mM NaOH aq., phosphate buffer (pH 7.4, 10 mM), 100 mM NaCl aq., and distilled water for 24 h each using a dialysis membrane. The solution was filtered with a 0.45-µm filter and freeze-dried to obtain tetra-PEG-GDL (yield: 900 mg). The obtained polymer was characterized by $^1$H-NMR using D$_2$O as the solvent.

**Preparation of PEG carriers**. PEG hydrogel, PEG sponge, and PEG slime were prepared by dissolving mutually cross-linkable tetra-PEGs and mixing the same volumes of the tetra-PEGs. These PEG carriers were crosslinked using the following precursors. (1) PEG hydrogel: tetra-PEG-PA and tetra-PEG-GS were separately dissolved in PB to give the concentration of PEG ($C_{PEG} = 60$ g l$^{-1}$). (2) PEG sponge: tetra-PEG-PA and tetra-PEG-GS were separately dissolved in PB with 250 mM K$_2$SO$_4$ to obtain $C_{PEG} = 60$ g l$^{-1}$. (3) PEG slime: tetra-PEG-GDL and tetra-PEG-FPBA were separately dissolved in PB to obtain $C_{PEG} = 60$ g l$^{-1}$.

**Microscopic observations of PEG carriers**. Red fluorescence-labeled tetra-PEG-PA and tetra-PEG-GDL were prepared to visualize the inner structure of the PEG scaffolds. For tetra-PEG-PA, 1000 mg of tetra-PEG-PA was dissolved in 20 ml of distilled water and stirred for 10 min at room temperature. Separately, 1 mg of Alexa-NHS was dissolved in 1 ml of super-dehydrated dimethyl sulfoxide and 16 µl (0.01 equiv. of tetra-PEG-PA) was added to the tetra-PEG-PA solution. The mixed solution was incubated at room temperature for 3 h, subsequently dialyzed against excess distilled water for 3 h to remove unreacted molecules, and freeze-dried to obtain tetra-PEG-PA labeled partly with red fluorescence, as a light-red powder (yield: 950 mg). The red fluorescence-labeled tetra-PEG-PA and tetra-PEG-GS were dissolved in PB with concentrations of K$_2$SO$_4$ = 0 and 300 mM to obtain $C_{PEG} = 60$ g l$^{-1}$. These two precursors were mixed in the same volume, poured into a cylindrical silicone mold (diameter: 5 mm and height: 1 mm), and incubated at 25 °C for 24 h. The obtained samples were observed using a CLSM.

For the synthesis of tetra-PEG-GDL, prior to the reaction of GDL with tetra-PEG-NH$_2$, 1000 mg of tetra-PEG-NH$_2$ was dissolved in 20 ml of super-dehydrated methanol. Sixteen microliters (0.01 equiv. of tetra-PEG-NH$_2$) of Alexa-NHS was then added to the solution and the solution was incubated for 24 h at room temperature. GDL and TEA were subsequently added to the mixed solution according to the method described for the preparation of tetra-PEG-GDL. Red fluorescence-labeled tetra-PEG-GDL and tetra-PEG-FPBA were then dissolved in PB to obtain $C_{PEG} = 60$ g l$^{-1}$. These two precursors were mixed in the same volume, poured into a cylindrical silicone mold, and incubated at 25 °C for 24 h, followed by observation using a CLSM.

**Fluorescence recovery after photobleaching (FRAP)**. Red fluorescence-labeled PEG precursors for PEG hydrogel and PEG slime were poured into a cylindrical silicone mold (diameter: 5 mm and height: 1 mm), and incubated at 25 °C for 24 h. Under CLSM, incubated PEG carries were bleached with a circle of 20 µm for 1 s, and the recovery of fluorescence intensity was observed.

**Dissolution of PEG carriers and the release behavior of model particles encapsulated in PEG carriers**. An in vivo model system, using a skin ulcer on the surface of mice backs, was designed with a 12-well cell culture insert (pore size of 8.0 µm) (Becton, Dickinson and Company, USA), which was placed on a 12-well culture plate, and the dissolution kinetics or release behavior was evaluated using this system. For the dissolution kinetics, 100 µl of a PEG scaffold labeled partly with red fluorescence at the end of the PEG molecule was prepared on the insert and incubated at 25 °C for 24 h. Then, 1000 and 2500 µl of DPBS was added to the insert and well plate, respectively. At room temperature in the dark, a 2500 µl sample from the well plate was collected to evaluate the concentration of permeated substances at each time point. After sampling the solution, a similar volume of fresh DPBS was added to the well plate to maintain the volume. The concentration of permeated samples was determined by measuring the fluorescence intensity ($\lambda_{ex} = 580$ nm, $\lambda_{em} = 590$ nm) using a microplate reader. The dissolution rate was

evaluated as a percentage compared with total amount of tetra-PEG labeled with red fluorescence.

To investigate the release behavior of a model virus, a solution of silica nanoparticles (1 mg ml$^{-1}$, sicastar$^\circledR$-green F, diameter of 30 nm, Micromod Partikeltechnologies, Germany) was suspended in each precursor solution of tetra-PEG-GDL for PEG slime and tetra-PEG-PA for PEG hydrogel or sponge at a concentration of 0.1 mg ml$^{-1}$ and $C_{PEG} = 60$ g l$^{-1}$. The mixed solution was added to the PEG precursors to prepare PEG carriers containing nanoparticles. The release behavior of the nanoparticles was evaluated according to the method described for the dissolution of PEG carriers, except for the measurement of the fluorescence intensity ($\lambda_{ex} = 460$ nm, $\lambda_{em} = 480$ nm).

**Stress relaxation test**. The stress relaxation test for PEG sponge was performed using the compression apparatus Rheogel-E4000 (UBM, Japan) for the cylindrical samples (diameter: 15 mm, height: 7 mm) at 25 °C. After the imposition of a small strain ($\varepsilon = 0.5$), the decay of stress was observed as a function of time. The relaxation of Young's modulus [$E(t)$] was calculated as stress divided by the applied strain. For PEG hydrogel and PEG slime, the dynamic viscoelastic measurements were performed using a stress-controlled rheometer (MCR301; Anton Paar, Graz, Austria) with a parallel plate fixture having a diameter of 25 mm. The angular frequency (0.1–100 rad s$^{-1}$) dependence of the storage ($G'$) and loss ($G''$) moduli were measured at 25 °C. The oscillatory shear strain amplitudes were confirmed within the range of the linear viscoelasticity for all the tests. We converted the $G'$ and $G''$ data to $E(t)$, assuming the isovolumetric deformation ($E = 3G$).

**Creation of AAV**. To generate the AAVs, plasmids of AAVDJ, pAAV-GFPNLS, pAAV-mCherryNLS, pAAV-DNP63A and pAD5 were prepared and transfected on cultured sub-confluent 293AAV, using the cesium chloride method[10]. The GFPNLS and mCherryNLS plasmid contained the nuclear localization signals (NLSs) sequence and the woodchuck hepatitis post-transcriptional regulatory element sequence. The number of gene copies was quantified using qPCR. The primer sequences that were used were as follows: AAV-ITR (Fw: GGAACCCC-TAGTAGTGATGGAGTT; Rv: CGGCCTCAGTGAGCGA).

**Preparation of PEG carriers containing AAV**. The AAV virus was diluted in PBS to make $5 \times 10^{10}$ gene copies (GC) in 5 µl. Approximately 5 µl of prepared AAV solution was mixed in 45 µl of tetra-PEG-GDL for PEG slime and tetra-PEG-PA for PEG hydrogel or sponge to $C_{PEG} = 60$ g l$^{-1}$, and the mixed solution was added to 50 µl of 60 g l$^{-1}$ PEG precursors to prepare PEG carriers containing AAV to obtain $C_{PEG} = 60$ g l$^{-1}$. Because PEG slime hardens immediately after mixing, the slime was formed into particle granules with a base of $1 \times 1$ cm and a size matching the ulcer base. For a control, 5 ul of PBS containing $5 \times 10^{10}$ GC of AAV virus was mixed in 95 ul of PBS.

**Ulcer creation and chamber attachments to mice**. Four-week-old female C57BL/6 mice were purchased from Nippon Bio-Supp (Japan). All animal experiments were approved by the Animal Research Committee of the University of Tokyo. The chambers made of silicon were synthesized by a 3D printer[20]. Briefly, a mold of a cap-shaped silicon chamber with a 5 mm wide flange was made using a 3D printer, and silicon was poured into the chamber by mixing two liquids. The synthesized silicon chamber was sterilized by autoclave.

The method of attachment was based on the method that we have previously reported for a hair reconstruction assay of skin ulceration[20]. The inter-scapular area was chosen for the chamber attachment site because of the high stability of the chamber fixation and the effectiveness in maintaining the ulcer surface over time. Under general anesthesia with isoflurane, the chamber attachment site was shaved thoroughly. Circular areas (1 cm in diameter) of the skin and subcutaneous tissue were surgically removed under the *panniculus carnosus*. The chamber was inserted, and the brim and overlying skin was sutured at 4 positions using 5-0 Ethilon$^\circledR$ (Johnson and Johnson, USA).

**Administration of AAV encapsuled in PEG carriers**. PEG hydrogel and PEG sponge formulations were administered by direct dropping of the liquid component onto the skin ulcer surface before the liquid hardened, through a small incision made near the top of the silicon chamber. After administration of the mixture, inhalation anesthesia was continued in the prone position for 5 min, and inhalation of the anesthetic was terminated after confirmation of curing at the ulcer surface. PBS containing AAV, the control group, was also administered in the same way, by dropping the liquid directly onto the skin ulcer surface in the chamber. PEG slime was administered in a different manner to the other materials because it was sufficiently hard to be grasped directly by forceps. The silicone chamber was opened 3/4 of the way around, and the slime was applied directly to the ulcer surface. After administration, the center of the chamber opening was sutured with a single stitch of 5-0 nylon to maintain a tight seal to avoid drying of the ulcer surface.

**Observation of ulcer surface and histological staining**. Mice were euthanized by cervical vertebral dislocation after inhalation anesthesia, the chamber was opened, and the back of the chamber was observed immediately after processing using a stereomicroscope (Axio Zoom$^\circledR$, Carl Zeiss, USA). Skin tissue, including the silicone chamber, was then carefully harvested up to just above the chest wall, including the midline muscle layer of the back, fixed overnight in 4% PFA, and placed overnight in 30% sucrose. Samples containing the fixed ulcer surface were embedded in optimal cutting temperature compound (O.C.T. Compound$^\circledR$, Sakura Finetek, JAPAN) and frozen specimens were prepared. Sections were prepared as 12-µm slices using a cryostat, washed twice with PBS, fixed in 4',6-diamidino-2-phenylindole (DAPI) mounting medium (Fluoromount-G$^\circledR$ with DAPI, Thermo Fisher Scientific, USA), and observed for fluorescence. Hematoxylin and eosin (HE) staining was performed with general technique[10].

**Cell counting**. From the images taken with a stereomicroscope in Axiozoom Z-stack mode, the slice with the most clearly discernible border of infected cells was used. For each specimen, five areas (20 µm square) with focus were randomly selected. The number of cells in the frame was visually counted and the mean value was calculated. The approximate number was calculated according to the area of the ulcer surface. Three sections obtained from three generally evenly distributed locations for each sample were extracted. Using the results from the HE staining of adjacent sections, the number of fluorescent particles or positive cells was visually counted under ×40 magnification field of view, divided into above and below the adipose tissue and muscle layers.

**Viral genomic quantification of liver samples**. To analyze the AAV titers for off-target effects in the liver, samples were collected from six lobes and genomic DNA was extracted using a commercial kit (DNeasy$^\circledR$ Blood & Sample Kit, QIAGEN, Germany). The copy numbers of AAV genome were measured on 100 ng of genomic DNA using qPCR on a StepOne Plus real-time PCR system (Thermo Fisher Scientific, USA). The primers used the viral genome sequence specifically for AAV-ITR (Fw: GGAACCCCTAGTAGTGATGGAGTT; Rv: CGGCCTCAGTGA GCGA) and the mouse titin gene (Fw: CTCCATCACTAGGGGTTCCT; Rv: TTCAGTCATGCTGCTAGCGC). The copy number of AAV genomes was expressed as an absolute value per diploid nucleus. All genomic DNA samples were analyzed in triplicate.

**Fluorescent labeling of AAV**. DNP63A-AAVDJ ($1 \times 10^{12}$ GC) was labeled using commercially available protein labeling kit (Alexa Fluor$^\circledR$ 568 Protein Labeling Kit, Thermo Fisher Scientific, USA) following the manufacturer's instructions[3].

**Isolation and culture of mouse adipose-derived stromal cells (mASCs)**. Subcutaneous groin-lumber fat pads were harvested from euthanized 3–5 weeks old mice. Adipose tissue was enzymatically digested[10] and the stromal vascular fraction was isolated by centrifugation, inoculated on a gelatin-coated 6-well plate using one well for each mouse specimen, and maintained in complete DMEM growth medium consisting of DMEM (containing 4.5 g l$^{-1}$ glucose, 110 mg l$^{-1}$ sodium pyruvate, and 4 mM L-glutamine) supplemented with 10% (v/v) heat-inactivated fetal bovine serum, 1:100 (v/v) MEM non-essential amino acid solution (Gibco), and 1:100 (v/v) GlutaMAX supplement (Gibco).

**In vitro tracking of labeled AAV**. P3 primary mASCs were dripped on a slide glass set on a cell culture plate, with $5 \times 10^4$ cells in 200 µl of complete DMEM growth medium. After 1 h of incubation in 37 °C, the medium was added to the plate up to the regular dose of cell culturing. After 24 h of incubation, MOI 2000 (GC per cells) of labeled AAVs were placed on the plate. Following 1 h of incubation at 4 °C, the cell attached to the cover glass was collected thoroughly and then mounted with DAPI stain to be assessed under a confocal microscope.

**In vivo tracking of labeled AAV**. Labeled AAV ($1 \times 10^{10}$ GC per mouse) and silica nanoparticles (final dose; 20 times dilution) were mixed in 100 µl of PBS and administered to the mice ulcer in the chamber. The ulcers were collected 24 h later for histological observation.

**Assessment of AAV decay over time**. P3 primary mASCs were seeded on a 24-well plate (base area 1.9 cm$^2$) with $1 \times 10^4$ cells in each well. After 24 h of incubation, the cells were administered GFPNLS-AAVDJ. GFPNLS-AAVDJ was incubated at 37 °C with or without mice blood serum (purified with BD Microtainer$^\circledR$ serum tube with separating gel) for a determined period (0–24 h) and then administered to mASCs. The culture medium was renewed 24 h after AAV administration. GFP-positive cells were counted 72 h from infection in quadruplicated wells.

**Statistics and reproducibility**. Excel software (Microsoft, USA) were used to assess statistical significance. To determine significance between two groups, unpaired two-tailed Student's $t$ tests were applied. $p \leq 0.05$ was considered as statistically significant. Data distribution was assumed to be normal. No statistical methods were used to predetermine sample size but our sample sizes were similar to those reported in a previous publication[10]. Data collection and analysis were not performed blindly.

**Reporting summary**. Further information on research design is available in the Nature Portfolio Reporting Summary linked to this article.

## Data availability
Source data for the graphs and charts in the figures is available as Supplementary Data and any remaining information can be obtained from the corresponding author upon reasonable request. Materials are available from the corresponding authors on reasonable request.

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

## Acknowledgements
This work was supported by JSPS KAKENHI Grant numbers JP20H03847 (to M.K.); JSPS KAKENHI Grant-in-Aid for Challenging Research (Pioneering) (JP20K20609) (to M.K.); JSPS KAKENHI Grant-in-Aid for Scientific Research (A) (JP21H04688) (to T.S.); Grant-in-Aid for Transformative Research Areas (JP20H05733) (to T.S.); Grants-in-Aid for JSPS Fellows (21J10828) (to M.K.); Grants-in-Aid for JSPS Fellows (JP20J01344) (to S.I.); Grants-in-Aid for Early Career Scientists (JP20K15338) (to T.K.); Japan Science and Technology Agency (JST) CREST under Grant number JPMJCR1992 (to T.S.); and AMED under Grant Number JP21zf0127002 (to M.O., T.S., and M.K.). Victoria Muir, PhD, from Edanz (https://jp.edanz.com/ac) edited a draft of this manuscript.

## Author contributions
T.S. and M. Kurita designed the study. M. Kato and S.I. designed the experiments. Data acquisition and/or analysis was performed by M. Kato, S.I., T.K., and T.N. AAVs were created by Q.S., Z.D., T.N., and M. Kurita. Polymers were synthesized by T.K. and M.N. M. Kato, S.I., T.K., T.S., and M. Kurita drafted the manuscript. Administrative, technical, or supervisory tasks were handled by M.O., T.S., and M. Kurita.

## Competing interests
The authors declare no competing interests.
