## [Peer Review File · Communications Biology]

Reviewers' comments:

Reviewer #1 (Remarks to the Author):

This manuscript analyzes the ability of biocompatible materials for localized gene transduction in skin ulcer animal models.

Increase of specificity in localized gene therapy with use of carriers is a very novel approach. The manuscript is well sounded and all the conclusions are properly supported by the data presented. I highly recommend publication of this manuscript.

A few minor points for the author to improve:

1. The authors mention one useful material as "dynamically crosslinked polymer" which might have a significant role for translation, but is not well explained. The authors should discuss this idea better and if they have any data to support it include it in the manuscript.
2. The data described in Fig 3b is unintelligible. The differences might be clear, but the technical details are needed to help the reader properly interpret the data.
3. Authors analyzed the efficiency of gene transduction and related to the frequency of positive-cells. I recommend the authors to try to see the efficiency not only with the frequency but also with the strength of expression.
4. Again, particle capacity of PEG on the surface seems very strong, but the efficiency of gene transduction efficiency in superficial layer is not enhanced in detail. What are the causes of this discrepancy? Proper discussion might help understand the readers the importance of this topic.

Reviewer #2 (Remarks to the Author):

Kato et al. reported a PEG-based carrier, named PEG slime, that enables localized recombinant adeno-associated virus (rAAV) delivery to mouse skin ulcer with limited off-target delivery. The findings are interesting and potentially useful for gene therapy applications that require localized gene delivery. The study was presented in a clear and logic manner. However, this reviewer has several major concerns:

1. The authors showed that PEG slime delivered rAAV specifically to the superficial layer, avoiding off-target to deeper tissues and liver (Fig. 3 and Fig. 4). What is the importance of gene delivery to superficial layer regarding skin ulcer therapeutics?
2. The entire study utilized fluorescence reporters as proof-of-concept, but would be more convincing if a therapeutic gene was employed to confer measurable skin ulcer therapeutic readouts.
3. Reporter nanoparticles were used for mechanistic studies (Fig. 5). Although the size of the nanoparticles (~ 30 nm) was carefully selected to be comparable to that of rAAV (~ 25 nm), using labeled rAAV particles would be a more straightforward and relevant experiment. There are various ways to label rAAV with a fluorophore in the literature.
4. Fig. 4b: the rAAV vector DNA in the liver was quantified to be 10^7 to 10^9 GC per diploid, which is far exceeding what has been described in the literature by any delivery method. Also, considering that 5×10^{10} GC of rAAV total was delivered, it is unlikely that the liver contains 10^9 GC per diploid.

Reviewer #3 (Remarks to the Author):

In the manuscript titled "In situ-formable, dynamic crosslinked poly (ethylene glycol) carrier for localized virus infection and reduced off-target effects" Kato et al explored the biomaterial carriers for topical therapeutic application of AAVs, especially for wound or ulcer healing. The authors

developed cross-linked PEG carriers and compared their physical, chemical, AAV retention and release properties in-vitro including in-situ mouse ulcer models. Stability of AAVs in carriers, release rates/kinetics and transduction efficiency in-situ are the crucial parameters to show efficacy of these biomaterials as carriers. Assessing any toxic or side-effects of these carriers for topical applications is also crucial for their use in therapeutics. There is a high unmet need for developing these sorts of biomaterials that could be applicable for AAV applications, therefore this work is of high importance to the community.

I have some comments and concerns that could help improve this manuscript

1. Title could be re-worded to flow better especially grammatically. Inclusion of the term 'AAV' could be better suited, as a generalized term 'virus infection' may be misleading to the broader audience.
2. Line 54 'AAV' instead of AVV
3. The kinetics and release rate of these carriers were thoroughly tested in-vitro using red fluorescent labelling but there is no data describing the stability, release rate or kinetics of AAVs either in cell lines in a dish or in-situ (on mouse ulcers).
4. Could authors clearly describe within result text or in methods section how these ulcers were created and what 'topical morbidity' is modelled here. Merely citing references in line 172 is not just enough. What are the dimensions of the ulcer
5. Why 72 hrs time frame for quantitative analysis was chosen? I'm not convinced with explanation in line 175-177 with reference. Authors could have generated data supporting this explanation with the carriers tested. Time-course data for AAV transduction efficiency in each biomaterial is needed to claim the highest effective efficiency with a few selected ratios of AAV: biomaterials.
6. In line 179-180 authors mentioned that "GFPNLA cells were more frequently observed in animals treated with AAVs...." Do you think this is the maximum achievable cell transduction? Could they provide dose-response curve for this data Fig3c?
7. For Fig 3d.f could authors provide the dimensions of ulcers and the dimensions of the spread of AAVs based on GFP positive cells? Also provide what layers in the dermis they mean by 'deep' and 'superficial'.
8. Authors should write/mention somewhere that AAV transduction efficiency was calculated based on 'number of GFP positive cells....'. Line 190 should have number of GFP positive cells instead of 'content' which is usually used if you are calculating the percentage of the GFP positive cells for a given ulcer. If you can provide transduction efficiency in cell percentages that would be better measure.
9. For lines 194-197 could be explained better if authors could comment on the distribution or spread of AAVs, do they think that PEG-slime treated animals have more localized high distribution in superficial layers than in deep layers when compared to PBS. And is this due to sustained or slow release of AAV particles and degradation kinetics of the slime.
10. Why did authors not check the AAV off target distribution in other organs other than liver especially kidneys?
11. For the result section "nanoparticles behavior on surface of skin ulcers", could authors provide the dose-dependency data for this distribution?
12. In the discussion section can authors include discussion about morphological/anatomical differences in human skin/dermis and mouse including some discussion on sweat glands, and AAV shedding dynamics.

We sincerely appreciate the informative comments from the reviewers. Below are the responses to each comment. The revised sentences in the manuscripts and figures are highlighted in red text.

Reviewer 1.

1. The authors mention one useful material as “dynamically crosslinked polymer” which might have a significant role for translation, but is not well explained. The authors should discuss this idea better and if they have any data to support it include it in the manuscript.

Response: We recognize that we have not included important information. To demonstrate that PEG slime is a dynamically crosslinked polymer, we observed fluorescence recovery after photobleaching (FRAP) of PEG slime and PEG hydrogel under a confocal laser microscope. The fluorescence intensity of PEG slime completely recovered approximately 60 s after bleaching, whereas that of PEG hydrogel was unchanged after bleaching. This indicated that PEG slime was a viscoelastic polymer liquid, in which crosslinking between PEG molecules was temporal.

Corresponding changes include the addition of Supplemental Fig. 1 and the following description in the main text:

“Fluorescence recovery after photobleaching (FRAP) measurements (**Supplemental Fig. 1**) showed that the fluorescence intensity of PEG slime was recovered, indicating that stress relaxation originated in the dynamic covalent bonds between GDL and FPBA. The fluorescence intensity of PEG slime was completely recovered approximately 60 s after photobleaching, whereas that of PEG hydrogel was unchanged. This confirmed the translational movement of the constitutive molecules of PEG slime and the dynamic crosslinks between these molecules.” (Line 120-127)

2. The data described in Fig 3b is unintelligible. The differences might be clear, but the technical details are needed to help the reader properly interpret the data.

Response: Thank you for the indication. To describe the technical details of our analyses, we revised Figure 3a in a more intelligible way. We also considered that an in-depth description of our models will be helpful and prepared Supplemental Fig. 2a.

Corresponding changes include the addition of Supplemental Fig. 2a and the following description in the main text:

“We surgically removed skin from the back of mice to generate an ulcer and isolated the resulting wound from the surrounding skin using a silicone chamber sutured to the deep fascia (19,20) (**Supplemental Fig 2a**). By isolating the wound from the surrounding skin using the chamber, we can study phenomena on the ulcer surface without interference from the surrounding skin, such as epithelialization and wound contraction.” (Line 182-188)

3. Authors analyzed the efficiency of gene transduction and related to the frequency of positive-cells. I recommend the authors to try to see the efficiency not only with the frequency but also with the strength of expression.

Response: Throughout the manuscript, we analyzed all fluorescence signals in binary measures (positive or negative) since the intensity of fluorescence signals in tissue samples are prone to technical artifacts, such as processing of tissues. However, we agree with the suggestion that the intensity of gene expression is another possible measure to evaluate gene transduction efficiency. Therefore, this time, we prepared green-fluorescent AAV (GFPNLS-AAVDJ) and red-fluorescent AAV (mCherryNLS-AAVDJ). Mixtures of these two vectors were administered on the surface of the ulcer and the frequency of duplicated gene transduction was measured. Unfortunately, the results were negative in terms of multiplicity of gene transduction (Supplemental Fig. 3). The number of mCherry-positive cells was smaller than that of GFP-positive cells, which likely resulted from the actual titer of the AAV solutions (the accuracy of qPCR-based AAV titering has technical restrictions) or sensitivity in different colors in the background signals. Nevertheless, we re-confirmed that the efficiency of gene transfer in the deep area was significantly lower than that in the superficial layer using both red-fluorescent and GFP-positive cells, and more importantly, we could not confirm that duplicated gene transduction in the slime group was enhanced compared with that of the PBS group. Although we could not detect an increase in the gene transduction efficiency in the superficial layer, this suggestion prompted us to provide complementary evidence.

Corresponding changes include the addition of Supplemental Fig. 3 and the following description in the main text:

“To further investigate the possible differences in gene transduction efficiency in terms of the multiplicity of gene transduction, mixtures of two different fluorescent colored AAVs (GFPNLS-AAVDJ and mCherryNLS-AAVDJ) were administered on the surface of the ulcer, and the frequency of duplicated gene transduction was measured (**Supplemental Fig. 3a**). The number of duplicated gene transduction was consistent between PBS and PEG slime, further indicating

that there were no significant differences in mice treated with PEG slime and PBS in terms of the multiplicity of gene transduction (**Supplemental Fig. 3b**).” (Line 221-229)

4. Again, particle capacity of PEG on the surface seems very strong, but the efficiency of gene transduction efficiency in superficial layer is not enhanced in detail. What are the causes of this discrepancy? Proper discussion might help understand the readers the importance of this topic.

Response: Thank you for important remark. We considered that this discrepancy, in which the intensity of fluorescent particles increased but AAV-mediated gene transduction in the superficial layer did not increase, might result from the chronological decay of AAVs in PEG. To elucidate the possible influences, we investigated the gene transducing ability of AAVs kept at 37 °C with or without mouse blood serum *in vitro*. The results suggested that when AAVs were kept 37 °C in the absence of serum, the gene transduction ability of AAV was gradually lost between 3 to 18 h. In contrast, when AAVs were kept for 24 h with serum, the gene transduction ability of AAV was maintained.

We considered that PEG slime might increase the absolute amount of AAVs administered in the superficial layer and that this increase was cancelled by the decay of AAVs within PEG over time, thereby reducing the number of positive cells in the deeper layers as well as the total number of positive cells.

Corresponding changes include the addition of Fig. 6 and the following description in the main text:

“With PEG slime, the number of silica nanoparticles in the superficial layer increased, although AAV-mediated gene transduction in the superficial layer did not increase. We considered that this discrepancy might result from the decay of AAVs in PEG over time.

To determine the possible effects of storage time at body temperature and solvent solution on the gene transducing ability of AAVs, we tested the gene transducing ability of AAVs kept for 0, 3, 6, 12, 18, and 24 h at 37 °C with or without mouse blood serum *in vitro* (**Fig. 6a**). When AAVs were kept at 37 °C for 3 to 18 h in the absence of serum, the gene transducing ability of AAVs gradually reduced to one tenth of the original ability. In contrast, when AAVs were kept for 24 h with serum, the gene transducing ability of AAVs was maintained (**Fig. 6b**). The gene transducing ability of AAVs within PEG slime, formulated with PEG and PBS, might reduce over time, whereas this reduction might be attenuated once AAVs were released from PEG slime. Over time, the decay of AAVs in PEG might contribute to the discrepancy in the silica nanoparticle behavior and gene transduction efficiency in the superficial layer.” (Line 277-292) and “PEG slime might increase

the absolute amount of AAV particle administered in the superficial layer, although this increase was cancelled by the decay of AAVs within PEG over time, thereby reducing the number of positive cells in deeper layers as well as the total number of positive cells.” (Line 379-382).

Reviewer #2:

1. The authors showed that PEG slime delivered rAAV specifically to the superficial layer, avoiding off-target to deeper tissues and liver (Fig. 3 and Fig. 4). What is the importance of gene delivery to superficial layer regarding skin ulcer therapeutics?

Response: We aimed to demonstrate the clinical applicability of our new technology, i.e., the generation of expandable epithelial tissues via the reprogramming of wound-resident mesenchymal cells, which enables all regions of the wound to re-epithelialize without the spatial constraints observed during normal healing.

During physiological wound healing, epidermal defects are repaired from the other epidermis. However, *in vivo* reprogramming allows *de novo* epithelialization and greatly enhances the capacity for the regeneration of cutaneous defects.

Transduction of reprogramming factors potentially induces direct reprogramming from non-epithelial cells to epithelial cells and, hence, the formation of ectopic epithelial tissues at sites

other than the superficial layer of skin ulcers. In addition, it is desirable to minimize potential adverse reactions that the transgene may induce in remote organs.

In our initial proof-of-concept study (Kurita et al. 2018;561:243-247.), no adverse reactions were detected. (In the trials after the publication of this study, hundreds of experiments were conducted without any signs of adverse reactions.) Nevertheless, we considered all possible efforts should be made for further development towards clinical applications.

We have been working on this type of development for our own purpose, although gene transduction methods that improve local specificity are important in all types of gene therapy development.

We agree that it would be easier to convey the importance of the described approach by specifically presenting our envisioned application, and therefore we have supplied the following description in the main text:

“We have described a new technology to generate expandable epithelial tissues via the direct reprogramming of wound-resident mesenchymal cells, which enables all regions of the wound to re-epithelialize without the spatial constraints observed during normal healing (9, 10). Transduction of reprogramming factors potentially induces direct reprogramming of non-epithelial cells to epithelial cells and hence the formation of ectopic epithelial tissues at sites other than the superficial layer of skin ulcers. In addition, it is desirable to minimize potential adverse reactions that the transgene may induce in remote organs.” (Line 387-394).

2. The entire study utilized fluorescence reporters as proof-of-concept, but would be more convincing if a therapeutic gene was employed to confer measurable skin ulcer therapeutic readouts.

Response:

We completely agree with the idea of showing measurable skin ulcer therapeutic readouts with the new technology.

However, as stated in the reply to comment 1, it is very difficult to show the benefit of reduced adverse event risks experimentally with our envisioned approach. The target morbidity could not be detected in the limited number of small animal studies. However, it is definitely

important for clinical applications.

We believe fluorescent reporters can provide potential solutions to clinically important problems, and thus they are important constituents of DDS formulations to enhance local specificity for clinical interventional applications.

Taking these factors into consideration, we have supplied the following description:

“In our initial proof-of-concept study, no adverse reactions were detected in the limited number of small animals. Nevertheless, all possible measures should be made for the further development towards clinical applications. We consider the current findings might be insightful not only for the development of gene therapy toward skin ulcers, but also for any types of therapeutic developments that are localized and thus expected to have extremely powerful effects.” (Line 396-401)

Thank you for your suggestions. The additional descriptions inspired by comments 1 and 2 will help the readers to understand the standpoint of the current study.

3. Reporter nanoparticles were used for mechanistic studies (Fig. 5). Although the size of the nanoparticles (~30 nm) was carefully selected to be comparable to that of rAAV (~25 nm), using labeled rAAV particles would be a more straightforward and relevant experiment. There are various ways to label rAAV with a fluorophore in the literature.

Response: Thank you for the instructive suggestion. We thoroughly investigated previous studies and found that multiple studies used fluorescent labels to evaluate the dynamics of the virus. We agree that tracing of labeled AAV particles is considered to be more straightforward for our purpose, and it would be of interest.

Among the various methods, protein-labeling of AAV particles is frequently used *in vitro* and *in vivo*, and therefore we tested protein-labelling method.

Consequently, we found labelled AAVs were useful in *in vitro* intracellular tracing (Supplemental Fig. 4a). However, the signals from labelled AAVs were insufficient for reliable detection, at least in our tissue samples of skin ulcer because the background signals were consistently high (Supplemental Fig. 4b, 4c).

For these reasons, at least in our evaluation of particle dynamics in the skin ulcer histological section, we considered fluorescent silica nanoparticle (Sicastar™, Micromod) was the better option, although there might be potential differences between AAV particles and fluorescent nanoparticles.

Corresponding changes include the addition of Supplemental Fig. 4 and the following descriptions in the main text:

“To elucidate the possible mechanistic properties specific to the PEG slime, fluorescently labeled AAVs (21-23) were tracked, and we found that reliable detection was difficult because of the weak fluorescence signals relative to the background signals (**Supplemental Fig. 4a-c**).” (Line 255-259) and “To investigate the detailed kinetics of AAVs, fluorescent labeling was used. However, reliable tracing was difficult because of the weak fluorescence signals relative to the background signals. Alternatively, we employed silica nanoparticle because each particle exhibited strong fluorescence and could be reliably traced in skin ulcer tissue samples. The possibility that AAV and silica nanoparticles may not exhibit the same behavior is a fundamental limitation of the current study.” (Line 301-306).

4. Fig. 4b: the rAAV vector DNA in the liver was quantified to be 10^7 to 10^9 GC per diploid, which is far exceeding what has been described in the literature by any delivery method. Also, considering that 5×10^{10} GC of rAAV total was delivered, it is unlikely that the liver contains 10^9 GC per diploid.

Response: Thank you for pointing this out. We are really sorry that we made a critical mistake during the calculations. This has been corrected in Fig. 4.

Reviewer #3

1. Title could be re-worded to flow better especially grammatically. Inclusion of the term ‘AAV’ could be better suited, as a generalized term ‘virus infection’ may be misleading to the broader audience.

Response: Thank you for your suggestion. We revised the title from “In situ-formable, dynamic crosslinked poly(ethylene glycol) carrier for localized virus infection and reduced off-target effects” to “**In situ-formable, dynamic crosslinked poly(ethylene glycol) carrier for localized adeno-associated virus infection and reduced off-target effects**” (Line 1-2).

2. Line 54 ‘AAV’ instead of AVV

Response: Thank you for pointing this out. We revised the mistyped letter, from AVV to AAV.
(Line 45)

3. The kinetics and release rate of these carriers were thoroughly tested *in-vitro* using red fluorescent labelling but there is no data describing the stability, release rate or kinetics of AAVs either in cell lines in a dish or *in-situ* (on mouse ulcers).

Response: Thank you for pointing out this important issue. Additional experiments were performed to determine AAV stability and kinetics.

To investigate the effect of incubation time on AAV stability, we evaluated the potency of GFPNLS-AAVDJ *in vitro*. When AAVs were incubated with the serum for 3–12 h at 37 °C, the gene transduction efficiency decreased (assuming that AAV remained in PEG on the ulcer), and when AAVs were incubated with the serum for 24 h at 37 °C, the potency reduced to less than 1/10 of that with the initial dose. In contrast, when AAVs were incubated with the serum, the gene transducing potency was maintained up to 24 h (assuming that AAVs remained in the tissue but were outside PEG). The result is summarized in Fig. 6.

For *in vivo* kinetics, additional experiments were performed to evaluate the actual localization of AAV particles using fluorescently labeled AAVs. We confirmed localization of labeled AAV particles *in vitro* (Supplemental Fig. 4a), although the detection was unreliable in tissue sections (*in vivo*) (Supplemental Fig. 4b, 4c).

We also considered the possibility of evaluating the kinetics of labeled AAV *in vitro* using our *in vitro* dissolution and diffusion system. However, the fluorescence from labeled AAVs was very weak, and it did not endure upon dilution for detection. The data below illustrate that labeled AAVs could be detected before it was diluted to less than 1/10, whereas the fluorescence of silica nanoparticles was attenuated in direct proportionality to the dilution factor up to 1000-fold.

In vitro assessment of the fluorescence intensity upon serial volumetric dilution. Concentration-dependent fluorescence is valid up to 1000-fold dilution for silica nanoparticles and up to 10-fold dilution for labeled AAVs.

We considered that the reviewer suggested additional experiments to explain our initial inconsistent findings between AAV transduction and silica nanoparticle distribution with PEG slime in that the silica nanoparticle distribution increased but AAV transduction efficiency was not enhanced in the superficial layer of ulcers.

To this end, the decay of AAVs in PEG over time (indicated by the evaluation of AAV stability over time) was considered one explanation, although we could not provide convincing evidence for the kinetics of AAV particles.

Corresponding changes include the addition of Supplemental Fig. 4 and following descriptions in the main text:

“To elucidate the possible mechanistic properties specific to the PEG slime, fluorescently labeled AAVs (21-23) were tracked, and we found that reliable detection was difficult because of the weak fluorescence signals relative to the background signals (**Supplemental Fig. 4a-c**.” (Line 255-259)

“To investigate the detailed kinetics of AAVs, fluorescent labeling was used. However, reliable tracing was difficult because of the weak fluorescence signals relative to the background signals. Alternatively, we employed silica nanoparticle because each particle exhibited strong fluorescence and could be reliably traced in skin ulcer tissue samples. The possibility that AAV

and silica nanoparticles may not exhibit the same behavior is a fundamental limitation of the current study." (Line 301-306).

4. Could authors clearly describe within result text or in methods section how these ulcers were created and what 'topical morbidity' is modelled here. Merely citing references in line 172 is not just enough. What are the dimensions of the ulcer

Response:

Thank you for bringing to our attention the inadequacy of the description of this issue. The model, which we called isolated skin ulcers, simulated the central portion of a large cutaneous ulcer. By isolating the wound from the surrounding skin using the chamber, we can investigate the phenomenon on the ulcer surface without interference from the surrounding skin, such as epithelialization and wound contraction. We agree with the need for additional information and made revisions accordingly.

Corresponding changes include revision of Fig. 3a, addition of Supplemental Fig. 2a, additional descriptions in the main text and method:

"To investigate the effects of the PEG carriers on gene transduction with an AAV for the treatment of skin ulcers, isolated skin ulcers that simulated the central portion of a large cutaneous ulcer were employed (10). We surgically removed skin from the back of mice to generate an ulcer and isolated the resulting wound from the surrounding skin using a silicone chamber sutured to the deep fascia (19,20) (**Supplemental Fig 2a**). By isolating the wound from the surrounding skin using the chamber, we can study phenomena on the ulcer surface without interference from the surrounding skin, such as epithelialization and wound contraction." (Line 178-188), and method as "Circular areas (1 cm in diameter) of the skin and subcutaneous tissue were surgically removed under the *panniculus carnosus*. The chamber was inserted, and the brim and overlying skin was sutured at 4 positions using 5-0 Ethilon[®] (Johnson and Johnson, USA)." (Line 590-594)

5. Why 72 hrs time frame for quantitative analysis was chosen? I'm not convinced with explanation in line 175-177 with reference. Authors could have generated data supporting this explanation with the carriers tested. Time-course data for AAV transduction efficiency in each biomaterial is needed to claim the highest effective efficiency with a few selected ratios of AAV: biomaterials.

Response: Thank you for your instructive comment. Before working on the experiment, we checked with rough observations of a small number of samples and chose this time for the analyses. Reliable/fine images could not be taken without killing the animals because of respiratory movement, and thus we did not conduct thorough analyses during the initial trials. With this comment, we obtained additional data for presentation in this revision.

GFPNLS-AAVDJ was mixed with the carriers and then inoculated on the ulcer. Stereoscopic images were obtained after 1, 2, 3, 5, 7, 9, 11, and 13 days ($n=3$ for each carrier). GFP expression levels on the ulcer surface reached sufficient intensity for detection on day 3, and relative GFP expression levels among different carriers were consistent up to day 13.

The data are supplied as Supplemental Fig. 2c. Together with the new data on the optimization of the viral titer, we have supplied the following description about the experimental setup in the main text:

“We tested different titers of the virus in PBS (Supplemental Fig. 2b) and observed the ulcer surface with a fluorescence microscope over time (up to day 13) (Supplemental Fig. 2c) and ulcers 72-h after administration of 10^{10} Gene copies (GC) of virus were selected for quantitative analysis (Fig. 3a).” (Line 195-200).

6. In line 179-180 authors mentioned that “GFPNLA cells were more frequently observed in animals treated with AAVs....” Do you think this is the maximum achievable cell transduction? Could they provide dose-response curve for this data Fig3c?

Response:

Thank you for the question. Similar to the time course assessment indicated in comment 5, we checked with rough observations of a small number of samples and chose the titer for this issue at the time of initial trials.

The titer of 10^{10} (GC/ ulcer) was employed because 10^9 was too weak, and reliable cell count of histological samples (already planned for later analyses) was difficult. In contrast, with 10^{11} , there were too many positive cells on the surface of the ulcer, the signals overlapped, and reliable count could not be achieved in stereoscopic analyses.

We agree that background information is helpful to the readers, especially possible

researchers who plan to use similar analyses in future, and thus we to obtained additional data for this issue.

The data are supplied as Supplemental Fig. 2b. Together with the new data on the optimization of the timeframe for quantitative analyses, we have supplied the following description about the experimental setup:

“We tested different titers of the virus in PBS (**Supplemental Fig. 2b**) and observed the ulcer surface with a fluorescence microscope over time (up to day 13) (**Supplemental Fig. 2c**) and ulcers 72-h after administration of 10^{10} Gene copies (GC) of virus were selected for quantitative analysis (**Fig. 3a**).” (Line 195-200).

7. For Fig 3d.f could authors provide the dimensions of ulcers and the dimensions of the spread of AAVs based on GFP positive cells? Also provide what layers in the dermis they mean by ‘deep’ and ‘superficial’.

Response:

We noticed that the description of our experimental model in the initial submission was insufficient for the readers. To describe the model clearly, we revised Fig. 3a for the general reader. To describe the model in detail, we prepared Supplemental Fig. 2a for readers who, thankfully, might be interested in our analyses. We believe these changes might help and we thank you for pointing this out.

8. Authors should write/mention somewhere that AAV transduction efficiency was calculated based on ‘number of GFP positive cells.....’ . Line 190 should have number of GFP positive cells instead of ‘content’ which is usually used if you are calculating the percentage of the GFP positive cells for a given ulcer. If you can provide transduction efficiency in cell percentages that would be better measure.

Response:

Thank you for your instructive comment. We changed “content” to “number” on line 213. We agree that cell percentages are more popular to describe transduction efficiency in tissues/organs with consistent structures and definite margins. In the current study on skin ulcer, there were no definite margins and heterogenous populations of cells, and thus we considered that the number of positive-cells rather than the proportion of cells was more

appropriate. It might be better for us to describe our reason, and thus we have supplied the following description of our reasoning in the main text:

“We employed the number of positive cells rather than the percentile because we considered that the former would be more appropriate for histological analyses of skin ulcer tissues, which consisted of heterogeneous types of cells with undefined margins.” (Line 190-194).

9. For lines 194-197 could be explained better if authors could comment on the distribution or spread of AAVs, do they think that PEG-slime treated animals have more localized high distribution in superficial layers than in deep layers when compared to PBS. And is this due to sustained or slow release of AAV particles and degradation kinetics of the slime.

Response:

Thank you for the helpful suggestion. With new data on AAV stability (comment 3), we are convinced that the distribution of both AAVs and nanoparticles in the superficial layer is higher with PEG slime than with PBS owing to the sustained/slow release and relative reduction of gene transduction efficiency in each layer (i.e., no change in the superficial layer and a decrease in the deep layers, which arises from the decay of AAVs in PEG over time).

Corresponding changes include the following description in the main text:

“PEG slime might increase the absolute amount of AAV particle administered in the superficial layer, although this increase was cancelled by the decay of AAVs within PEG over time, thereby reducing the number of positive cells in deeper layers as well as the total number of positive cells.” (Line 379-382).

10. Why did authors not check the AAV off target distribution in other organs other than liver especially kidneys?

Response:

In our previous study, we thoroughly investigated the distribution of AAVDJ in various organ tissues using the IVIS Imaging system for detection. Below is the panel from the corresponding paper (Extended data Figure 3m from Kurita et al. 2018) showing *in vivo* luminescence images 1 week after the administration of luciferase-expressing AAVs through tail vein injection, interscapular subcutaneous injection, or inoculation in a chamber on the back (B, brain; Ce, caecum; Co, colon; Es + St, esophagus and stomach; Ey, eye; F, fat; H, heart; K + Ad, kidney and adrenal gland; L, liver; Lu, lung; Ov + U, ovary and uterus; Pa,

pancreas; SI, small intestine; Sk, skin; Sp, spleen).

Systemic directivity of AAVDJ is mainly focused on the liver, and therefore we checked the liver as the main distant off-target organ in the current study. We agree that the description in the initial manuscript was unclear and have provided the following description of our reasoning in the main text:

“AAV cell tropism differs among serotypes, and thus the importance of each organ as the distant off-target depends on the serotype. Previously, we assessed the distribution of AAVDJ in various organ tissues following the injection of luciferase-AAVDJs and found that luciferase expression is primarily confined to the liver (10).” (Line 236-240).

11. For the result section “nanoparticles behavior on surface of skin ulcers”, could authors provide the dose-dependency data for this distribution?

Response:

Thank you for your suggestion. We agree that analyses of dose dependency are always of interest to the readers. As we are looking for clinical applications of PEG as a drug delivery carrier (not only for AAV), these analyses would also be interesting to us. For a solid understanding of drug delivery in the subcutaneous region of the skin, where details are not yet available, more analyses will need to be carried out because time-dependent local

distributions of administered substances are influenced not only by simple diffusion but also absorption via lymphatic and peripheral vessels as well as clearance from systemic circulation. We are now trying to analyze these factors thoroughly. As reference, we would like to present a recent study (ACS macro letter, in revision, preprint information below*), which analyses the molecular weight-dependent diffusion, biodistribution, and clearance of tetra-armed poly(ethylene glycol) subcutaneously injected into the back of mice.

In the current study, “nanoparticles behavior on surface of skin ulcers” was investigated to determine the effect of PEG carriers on the ulcer and we considered that the use of PEG carriers (especially PEG slime) prolonged the residency time of nanoparticles on the surface was adequate for our purpose.

Taking all these factors into consideration, we have provided the following description of our reasoning:

“The behavior of silica nanoparticles on the surface of skin ulcers with or without PEG carriers indicated the potential utility of PEG carriers in the delivery of other types of drugs and biomolecules. Elucidation of other behaviors, such as dose dependency along with time-dependent diffusion, biodistribution, and clearance of subcutaneously administered substances should be performed in future studies.” (Line 372-377).

*Preprint information

MS ID#: BIORXIV/2023/531818

Molecular Weight-dependent Diffusion, Biodistribution, and Clearance Behavior of Tetra-armed Poly(ethylene glycol) Subcutaneously Injected into the Back of Mice

Shohei Ishikawa, Motoi Kato, Jinyan Si, Lin Chen Yu, Kohei Kimura, Takuya Katashima, Mitsuru Naito, Masakazu Kurita, and Takamasa Sakai

12. In the discussion section can authors include discussion about morphological/anatomical differences in human skin/dermis and mouse including some discussion on sweat glands, and AAV shedding dynamics.

Response:

Thank you for your suggestion regarding the limitations of the current study, which we have addressed with the following description in the main text:

“The other limitations of the current study lie in the differences between human and mouse skin

structures (31). The major differences include the thickness of layered components, such as the dermis and subcutaneous adipose tissue, the absence of sweat glands in mice, and the presence of the thin muscular layer known as *Panniculus carnosus* in mice back skin. These differences might not substantially influence the findings of the current study because the experiments were performed with the skin ulcer in a silicone chamber. Nevertheless, careful interpretation of the results is required during clinical applications. The other limitation lies in the experimental model. Administration of liquid AAVs in a closed chamber is completely different from actual administrations in clinical settings, which involve direct application of the solution to the open wound surface. Use of viscoelastic liquids as a carrier of AAVs might be advantageous for reducing AAV shedding (32), although it could not be determined experimentally." (Line 410-423).

REVIEWERS' COMMENTS:

Reviewer #1 (Remarks to the Author):

The authors have addressed all my concerns and this manuscript is now suitable for publication.

Reviewer #2 (Remarks to the Author):

The authors addressed my concerns by including helpful discussions. For the suggested experiments, the authors acknowledged their importance and/or relevance, but were not able to perform due to technical difficulties that were well explained in the revised text. I have no more questions.

Reviewer #3 (Remarks to the Author):

The authors have addressed my concerns satisfactorily in the revised version of the manuscript.